# Journey of Rosmarinic Acid as Biomedicine to Nano-Biomedicine for Treating Cancer: Current Strategies and Future Perspectives

**DOI:** 10.3390/pharmaceutics14112401

**Published:** 2022-11-07

**Authors:** Motamarri Venkata Naga Lalitha Chaitanya, Arya Kadukkattil Ramanunny, Malakapogu Ravindra Babu, Monica Gulati, Sukriti Vishwas, Thakur Gurjeet Singh, Dinesh Kumar Chellappan, Jon Adams, Kamal Dua, Sachin Kumar Singh

**Affiliations:** 1School of Pharmaceutical Sciences, Lovely Professional University, Phagwara 144411, Punjab, India; 2Faculty of Health, Australian Research Centre in Complementary and Integrative Medicine, University Technology Sydney, Ultimo, NSW 2007, Australia; 3Chitkara College of Pharmacy, Chitkara University, Rajpura 140401, Punjab, India; 4School of Pharmacy, International Medical University, Bukit Jalil, Kuala Lumpur 57000, Malaysia; 5Discipline of Pharmacy, Graduate School of Health, University of Technology Sydney, Ultimo, NSW 2007, Australia

**Keywords:** rosmarinic acid, cancer, novel drug delivery systems, clinical translation

## Abstract

Rosmarinic acid (RA) is a polyphenolic metabolite found in various culinary, dietary sources, and medicinal plants like *Coleus scutellarioides* (Linn) Benth., *Lavandula angustifolia* Linn., *Mellisa officinalis* Linn., *Origanum vulgare* Linn., *Rosmarinus officinalis* Linn., *Zataria multiflora* Boiss. and *Zhumeria majdae* Rech. F. Apart from its dietary and therapeutic values, RA is an important anticancer phytochemical owing to its multi-targeting anticancer mechanism. These properties provide a scope for RA’s therapeutic uses beyond its traditional use as a dietary source. However, its oral bioavailability is limited due to its poor solubility and permeability. This impedes its efficacy in treating cancer. Indeed, in recent years, tremendous efforts have been put towards the development of nanoformulations of RA for treating cancer. However, this research is in its initial stage as bringing a nanoparticle into the market itself is associated with many issues such as stability, toxicity, and scale-up issues. Considering these pitfalls during formulation development and overcoming them would surely provide a new face to RA as a nanomedicine to treat cancer. A literature search was conducted to systematically review the various biological sources, extraction techniques, and anticancer mechanisms through which RA showed multiple therapeutic effects. Various nanocarriers of RA pertaining to its anticancer activity are also discussed in this review.

## 1. Introduction

Cancer is one of the world’s major causes of deaths and is considered a global disease that is characterized by the uncontrolled proliferation of cells. The major causes of cancer include both exogenous and endogenous risk factors. Among the exogenous risk factors, an unbalanced diet, smoking, and chronic inflammation are the major causes of cancer at an early age. However, in the later stage of life, cancer is caused by the endogenous process that results in the formation of oxidant by-products after normal metabolism. These oxidants cause oxidative damage to DNA, proteins, and lipids, which leads to cancer [1].

As per the cancer statistics of 2021, 27% of men are diagnosed with prostate cancer, followed by lung, bronchus, and colorectal cancer (CRC). In the US, one-third of women are diagnosed with breast cancer, followed by lung cancer and CRC [2]. Worldwide, in every 6 deaths, 1 accounts due to cancer [3]. About 16.2 million cancer-related deaths and 28.0 million new cancer cases are expected worldwide by 2040 [4]. Various types of cancers with an estimated number of new cases and deaths by the end of the year 2022 are depicted in Table 1. About 11.7% of cases of breast, 11.4% cases of lung, 10% cases of CRC, 7.3% cases of prostate, 5.6% cases of stomach, 4.7% cases of liver, 3.1% cases of oesophagus, and 3.1% cases of cervical cancer cases were reported in 2021 according to the report from the World Health Organization (WHO). On the other hand, about 8% of cases of lung, 9.4% of cases of colorectal, 8.3% of cases of liver, 7.7% of cases of stomach, 6.9% of cases of breast, 5.5% of cases of oesophagus, 4.7% of cases of pancreas, and 3.8% of cases of prostate cancer lead to mortality [4]. In 2021, the percentage incidence of cancer cases throughout the world has been found as about 49.3% in Asia, 22.8% in Europe, 13.3% in North America, 7.6% in Latin America, and 5.7% in Africa [5].

Considering the frequency and emergence of new cases, substantial advancements have been made in cancer treatment, which include surgical procedures, radiation therapy, chemotherapy, targeted approaches, and hormonal procedures [6]. However, each of these modern therapies has its own side effects, which limits their usage [7]. Hence, the search for a cost effective, safe, and effective treatment with less side effects to treat cancer is an ongoing process. This search for novel, safe, and effective therapeutics has led the research toward utilization of herbal drugs and their isolated components to treat cancer. Considering the safety aspects of these phytopharmaceuticals and their anticancer effect through multiple pharmacological pathways, the herbal and dietary supplements have been utilized to improve the epigenetic mechanism to inhibit the spread of cancer.

Numerous epidemiological studies have been reported about the increased risk of cancer’s incidence in populations with inadequate intake of herbs [8], fruits [9], and vegetables [10]. Therefore, the inclusion of the same into the diet contributes to various pharmacological actions. Due to the presence of various bioactives, these herbal drugs may reduce the incidence of cancer [11]. Hence, exploration of plant bioactives for the treatment of cancer leads to the development of exciting new drugs.

The therapeutic potential of the isolated drug leads, as well as their structural-activity connections, are being investigated for possible inclusion in future molecularly-tailored anticancer medicines [12]. Thus, natural compounds with broad structural and chemical diversity, therapeutic potential, and multiple-targeting mechanisms are receiving increased attention in the quest for new medications [13,14].

The botanicals with chemo-preventive actions are primarily composed of polyphenols, for example, quercetin isolated from *Allium cepa* Linn (Liliaceae), genistein isolated from *Glycine max* Linn (Fabaceae), resveratrol extracted from *Vitis vinifera* Linn (Vitaceae), curcumin from *Curcuma longa* Linn (Zingiberaceae), RA from *Rosmarinus officinalis* Linn (Lamiaceae), etc. [15]. It is pertinent to add here that cancers of the breast, prostate, and digestive tract are less common in Asian populations, owing to their daily dietary habits of consumption of rich flavonoids and phenols [16].

Among polyphenols, rosmarinic acid (RA) has gained the attention of researchers among various phytoconstitutents owing to its chemopreventive action. *Rosmarinus officinalis* Linn, which is commonly known as rosemary belongs to the Lamiaceae family. It contains a potent polyphenolic compound RA [17]. In Indian [18] and African [19] traditional folk medicine, the plant has been used in the promotion of hair growth, reduction of renal colic pain, treatment of dysmenorrhoea, and soothing of bronchial problems [20]. In addition to the aforementioned uses, it has been used as a flavouring agent in foodstuff and beverages. Further, it has been found that the plant is useful in cosmetics [21]. Rosemary extract-containing supplements have been reported for the presence of choleretic, hepatoprotective, and antitumorigenic properties [22].

The molecular formula of RA is C_18_H_16_O_8_ and contains caffeic acid, tyrosine, and dihydroxyphenylalanine, which contribute to its molecular mass of 360 Da [23]. The structure of RA contains two benzene rings that are joined by two ortho-hydroxy groups at each end, and the structure is shown in Figure 1.

Chemically, RA is an ester of caffeic acid and 3,4-dihydroxyphenyl lactic acid which is predominantly found in plants belonging to the Lamiaceae and Boraginaceae family [24]. RA is present in many plant species like *Coleus scutellarioides* (Linn) Benth., *Lavandula angustifolia* Linn., *Mellisa officinalis* Linn., Mentha sp., *Origanum vulgare* Linn., *Rosmarinus officinalis* Linn., *Salvia* sp., *Satureja* sp., *Thymus* sp., *Zataria multiflora* Boiss., and *Zhumeria majdae* Rech. F that belong to Lamiaceae and species like *Rindera graeca* (A.DC.) Boiss. &Heldr, *Borago officinalis* Linn, *Symphytum officinale* Linn, *Sarcandra glabra* Linn thatbelong to the boraginaceae family [24].

The extensive literature has shown the therapeutic potential of RA as antibacterial [25], antiviral [26], anti-inflammatory [27], anti-cancer [28], immunological modulation [29], health-enhancing [22], and containing antioxidant properties [30]. Among these activities, a significant number of reports have highlighted the anticancer activity of RA [31,32,33,34,35]. Hence, it is important to investigate the role of RA as an anticancer lead which might further lead to the discovery of a new cytotoxic drug. The current systematic review describes the various extraction procedures of RA, significance of RA in various cancers, and challenges in the clinical translation of RA into cancer therapy. Further, the limitations of the clinical translation of RA and the role of novel drug delivery systems (NDDS) of RA against cancer has been covered.

## 2. Extraction Processes of RA

Various types of RA extraction techniques have been reported, which are discussed below.

Su et al. extracted RA by aqueous enzymatic extraction (EE) with the help of a variety of enzymes, such as cellulase and proteases from the dried leaves of *Melissa officinalis*.The amount of RA collected (mg/g of enzyme) by the use of different enzymes has been found as flavourzyme 16.22 ± 0.08, Chamzyme 16.77 ± 0.20, Protamex 20.96 ± 0.04, Cellulase A 21.88 ± 0.03b, Bromelain 20.46 ± 0.33, Papain 16.13 ± 0.30, Cellulase A + Protamex (1:1, *w*/*w*) 23.27 ± 0.10, and no enzyme (water only) 14.33 ± 0.19. Therefore, it was clear that the amount of RA extracted is more when in the presence of enzymes than without enzymes. A thin layer chromatography (TLC) technique was used to determine RA. Further, they have reported that EE is a green method which does not require any organic solvent and can be used in the presence of aqueous solvents for the extraction of RA. The extraction process worked best with a Cellulase A and Protamex mixture (1:1, *w*/*w*) out of all the tested enzymes. With the help response surface methodology (RSM), a maximum RA concentration of 28.23 ± 0.41 mg/g was attained with an enzyme loading of 4.49%, a water-to-sample ratio of 25.76 mL/g, a temperature of 54.3 °C, and a 2-h extraction period [36]. Figure 2 shows various extraction methods of RA.

Thungamunnithum et al. extracted RA by the use of green ultrasound assisted extraction (USAE) from *Plectranthus scutellarioides* Linn., which is an ornamental plant. They used an ultrasonic bath USC1200TH (Prolabo, Sion, Switzerland) with an inner dimension of 300 × 240 × 200 mm. USAE requires an electrical power of 400 W (i.e., acoustic power of 1 W/cm^2^), digital timer, and a temperature controller. The sample was suspended in 20 mL of aqueous ethanol and put in a 100-mL quartz tube with a vapour condenser on top. Various aqueous ethanol concentrations (0, 25, 50, 75, and 100% (*v*/*v*)) and solvent to material (S/M) ratios (10:1, 25:1, and 50:1) were investigated. Further, different extraction temperatures (25, 35, 45, 55, 65, and 75 °C), ultrasonic frequencies (0, 15, 30, and 45 kHz), and extraction times (0, 15, 30, 45, 60, and 75 min) were also examined. After centrifuging the extract for 15 min at 3000 rpm, the supernatant was filtered (0.45 m) for HPLC analysis. This process of extraction gave a clear indication that ethanol is a green extraction solvent in an ultrasonic bath; the *P. scutellarioides* leaves had a maximum RA of 11.08%. The proposed approach was shown to be more effective and quicker than the traditional heat reflux approach. It was determined through comparison with RA yields from several *P. scutellarioides* systems that the leaves of this ornamental plant can serve as a raw material for an effective extraction of RA [37].

Despite the fact that there are many studies on the extraction of RA from plants, the conditions for extraction and the impact of the key factors on RA output remain unpredictable. Sik et al. researched on six different Lamiaceae plants (rosemary, oregano, sage, peppermint, and lemon balm) to identify the conditions and factors governing the RA extraction. They have used both conventional and nonconventional extraction to identify the optimal extraction process. A variety of extraction methods including maceration with stirring (MACS), heat reflux (HRE), and microwave-assisted extraction (MAE) with various process conditions like solvent acidity, solvent type, extraction time, and temperature have been considered. Under test conditions, high-performance liquid chromatography with diode-array detection (HPLC-DAD) was used to measure the RA content. According to their findings, compared to other solvent systems, acidified aqueous ethanol extraction (EtOH-H2O-HCl, 70:29:1, *v*/*v*/*v*) was the most effective method for recovering RA [38].

Liu et al. extracted RA with a microwave-assisted method with the application of ionic liquids. For this, 2.45 GHz of microwave radiation was broadcasted from a microwave oven. The oven’s maximum output power was 700 W. The entire apparatus was operated at atmospheric pressure. Ionic liquids were added, and this significantly increased the extraction yields of RA. In this work, the extraction yield of a variety of 1-alkyl-3-methylimidazolium ionic liquids with varying cation and anion compositions was studied. The composition of ionic liquids, particularly the cations, has a big impact on how much of the desired constituents can be extracted. The ideal circumstances for this strategy were looked into and progressively determined. Investigations were also conducted on the microstructures and chemical composition of the samples of rosemary, both before and after extraction. Furthermore, the stability, repeatability, and recovery experiments supported the suggested method’s validity. According to the findings, the developed method offered a good substitute for both the extraction of RA and essential oils (EO) from rosemary and other herbs. The ideal process needed less time for extraction compared to conventional extraction methods and produced greater extraction yields of RA, which was 3.97 mg/g [39].

## 3. Role of RA in Various Types of Cancer: Emphasis on Their Mechanism of Action

RA has been reported to have anti-carcinogenic effects against various cancers [40,41,42,43]. A detailed discussion about the mechanisms of action of RA against each cancer has been discussed under this section.

### 3.1. Oral Cancer

Luo et al. reported the anticancer potential of RA in oral cancer cell lines (SCC-15). The dose-dependent anti-proliferative activity of RA was reported in SCC-15 cancer cells through various mechanisms. The inhibition of oncogene cyclin-dependent kinase (CDK)-8 resulted in apoptosis induction and also caused cell cycle arrest at the G2/M phase of the SCC-15 cancer cells. Treatment with RA also induced stress in the endoplasmic reticulum which negatively affected the migratory potential of the cancer cell [44].

Anusuya and Manoharan studied the effect of RA on 7, 12-dimethylbenz (a) anthracene (DMBA) induced oral carcinogenesis in Hamster buccal pouch tissue. Oral carcinogenesis in a Golden Syrian Hamster was induced by exposing the buccal pouches to 0.5% DMBA in liquid paraffin 3 times a week for 14 weeks. After 14 weeks, histopathology of buccal pouches confirmed squamous cell carcinomas in hamsters. Oral administration of RA (100 mg/kg body weight) was found to inhibit tumour growth in DMBA-treated hamsters. It was also observed that the biochemical and molecular markers in DMBA-treated hamsters were found to be significantly enhanced after RA administration. Hence, the study findings reported that RA suppressed oral carcinogenesis by enhancing detoxification enzyme activity, decreasing lipid peroxidation, improving antioxidant status, and down-regulating the expression of Bcl-2 and p53 in DMBA-induced oral carcinogenesis.

### 3.2. Colorectal Cancer (CRC)

RA has shown potent activity against metastasis in CRC cells through the activation and phosphorylation of 5΄adenosine monophosphate-activated protein kinase (AMPK). The anti-proliferative effect of RA was studied in murine colon carcinoma colon 26 cells (CT26) and human colon carcinoma cell lines (HCT116). Time and concentration-dependent inhibition in cell proliferation was observed. Further, induction of apoptosis and cell cycle arrest in the G0/G1 phase has been observed.

It has been observed that upon treatment with RA, invasion and migration of CRC cells are inhibited as well as matrix metalloproteinase (MMP) expression; MMP-2 and MMP-9 were decreased especially. In addition to these, the expression of adhesion and adhesion molecules such as ICAM-1 and integrin β1 in CRC cells were decreased after its treatment with RA [31].

RA has multiple modes of action on colitis-related malignancies. Hence, the anti-tumour activity of RA and its molecular mechanisms were investigated in conditioned culture media (in vitro studies), and Azoxymethane (AOM)/Dextran.Sodium Sulfate (DSS) induced colitis-associated colon cancer (CAC) in a murine model (in vivo studies). It was observed that RA reduced the severity of colitis, decreased expression of proteins that are related to inflammation, decreased tumour incidence, and blocked the formation of colonic adenoma in the CAC induced murine model. It was observed that RA also lowered the anti-apoptotic factors that promoted tumour development and suppressed toll-like receptor 4 (TLR-4) in the inflammatory condition. RA caused significant modulation of TLR-4 mediated nuclear factor- kappa B (NF-KB) and the signal transducer and activator of transcription 3 (STAT3) in a pleiotropic manner that resulted in attenuation of anti-apoptotic factor. In vitro studies confirmed the inhibition of CM-induced TLR4 overexpression as well as competitive inhibition of TLR-4 myeloid differentiation. These studies confirmed that RA reduces the growth of human colon cancer cells and inhibits the progression of the disease in an inflamed milieu [33].

Another, driving factor for the development of tumours is the “Warburg effect”. It is a metabolic phenotypic condition exhibited by cancer cells that are predominantly glycolytic in nature, even when oxygen is plenty. Yichun et al. found that RA at varying concentrations (33, 75, and 100 mol/L) has exhibited an anti-Warburg effect activity against CRC. It was observed that after treatment with RA, CRC cells consumed less glucose and produced less lactate. Along with these effects, RA inhibited transcription factor hypoxia-inducible factor-1 (HIF-1), which is a part of the glycolytic pathway [45].

In another study, the varying concentrations of RA (5, 10, 20 µmol/L) reduced cyclooxygenase(COX)- promotor activity induced by 12-O-tetradecanoylphorbol-13-acetate (TPA) in colon cancer cell lines (HT-29) as well as reduced TPA-induced transcription of the activator protein-1 (AP-1) promotor-luciferase construct. These observations confirmed the antagonizing effect of RA on AP-1-dependent activation of COX-2 expressions. In addition to this RA also antagonized the activation of extracellular signal-regulated protein kinase-1/2 (ERK1/2) [46].

In vitro studies in colorectal adenocarcinoma cell lines (Ls174-T) with varying doses of RA from 0 to 80 µg/mL reported more than 70% cell viability. This indicated the ability of RA to induce metastasis without cytotoxicity. Further assays and studies were performed in the Ls174-T cells and reported the dose-dependent inhibition of migration, adhesion, and invasion of cells. RA treated cells also reported a concentration-dependent decrease in reactive oxygen species (ROS) content, decreased gelatinase activity, decreased translocation of NF-κB, down-regulation of p-Akt and p-ERK, and inhibited expression of VEGF, MMP-2, and MMP-9. A significant inhibition in the formation of metastasis nodules was obtained at each selected dose. However, a slight decrease in weight loss has been reported at a dose of 4 mg/kg, which shows the possibility of cytotoxicity in mice. Therefore, a dose of 2 mg/kg was recommended for suppression of tumour growth in the in vivo mice model [47].

Another in vivo study was carried out in Wistar rat colon carcinoma was induced in groups 3 to 6 groups by subcutaneous injection of 1,2-dimethylhydrazine (DMH) once weekly for the first four weeks. After induction, RA was administered orally to 4, 5, and 6 groups at a dose of 2.5, 5, and 10 mg/kg body weight, respectively. Tumour formation after DMH induction was confirmed by the formation of aberrant crypt foci and their multiplicity in the colon. Among the three doses of RA, the group supplemented with 5 mg/kg body weight has shown a more pronounced effect in decreasing the aberrant crypt foci by approximately 50% in comparison to the DMH-induced group. A significant alteration in the antioxidant status and oxidative stress markers (superoxide dismutase, catalase, glutathione peroxidase) was observed. Further, the activities of faecal and colonic bacterial enzymes such as β-glucuronidase, β-glucosidase, mucinase, nitroreductase, and sulphatase were reported to be decreased after RA administration [48].

### 3.3. Ovarian Cancer

Anti-proliferative activity of RA in human ovarian (OVCAR-3) cancer cells was evaluated using a 3-(4,5-dimethylthiazol-2-yl)2,5-diphenyl tetrazolium bromide (MTT) assay. A time and concentration-dependent cytotoxic effect was observed, and IC_50_ values at 48 h and 72 h were found to be 34.6 and 25.1 µM, respectively. Further, scanning electron microscopy (SEM) analysis confirmed a significant alteration in the cell surface morphology such as cell shrinkage and rounding, membrane blebbing, the disappearance of the microvillious structure, and the formation of condensed apoptotic bodies. Further, RT-PCR indicated a dose-dependent increase in the expression level of lncRNA MALAT-1 in OVCAR-3 cancer cells after treatment with RA. These observed changes in the OVCAR-3 cancer cells after treatment with RA confirmed its anticancer activity induced by apoptosis, inhibition of cell migration, and modulation through lncRNA MALAT-1 expression [49].

### 3.4. Cervical Cancer

The mTOR (mammalian Target of Rapamycin 1)/S6K1 (ribosomal protein S6 kinase 1) signalling pathway was found to be typically active in cervical cancer cells and acts as a biological target during treatment. In combination with conventional chemotherapy, RA methyl ester (RAME) was found to synergistically inhibit mTOR/S6K1 in cervical cancer cell lines (Henrietta Lacks). Virtual screening of RAME confirmed that its interaction with mTOR is followed by the inhibition of S6K1. Unlike RA, RAME triggered autophagy and apoptosis, which decreased cell survival. In this study, cisplatin-resistant cervical cancer cells were treated with RAME as well as cisplatin. Cancer cells treated with RAME resulted in a considerable increase in activity, and this was most likely due to mTOR/S6K1 inhibition-mediated autophagy and apoptosis [42]. The detailed mechanism of Ra has been explained in the Figure 3.

### 3.5. Lung Epithelial Cancer

RA rich fraction (RA-RF) has been isolated from the seed meal of *Perilla frutescens*. This plant was found to reduce oxidative stress, inflammation, and cancer cell spread on PMFF (Particulate matter from forest fires) induced A549 (human lung epithelial cells). In the study, RA-RF dramatically reduced ROS production as well as decreased the mRNA expression of interleukin (IL)-6, IL-8, tumour necrosis factor (TNF-α), and cyclooxygenase-2 (COX-2). RA-RF also suppressed the metastatic cascade, affecting matrix metalloproteinase-9 (MMP-9) activity and decreasing cancer cell invasion and migration via the AP1 (Activator protein 1), Nuclear factor kappa B NF-κB, and Protein kinase B (Akt) signalling pathways. The molecular mechanism of RA in regulating NF-κB has been clearly explained in the following Figure 4.

### 3.6. Non-Small Cell Lung Cancer (NSCLC)

Cis-diaminedichloroplatinum(II) (DDP) is a first-line chemotherapeutic agent used in the treatment of non-small cell lung cancer (NSCLC). However, repeated administration of cisplatin to the patients reported chemoresistance. Liao et al. investigated the ability of RA to effectively reverse multi-drug resistance in NSCLC. In vitro studies were carried out in adenocarcinoma human alveolar basal epithelial cell lines (A549) and A549DDP (A549 modified cisplatin-resistant) cell lines. The results of in vitro studies in these cell lines suggested dose-dependent inhibition in NSCLC cell proliferation after the application of a combined treatment with RA and DDP. In addition, a synergistic effect was reported in the induction of cell cycle arrest at the G1 phase.

Induction of apoptosis occurs due to the activation of JNK phosphorylation: mitochondria-mediated apoptosis, activation of caspase, and increased sensitivity of A549DDP cell lines.

In order to verify the mechanism of drug resistance in A549DDP cell lines, quantification of MDR1 mRNA and P-gp protein levels was carried out. The results suggested that in comparison to normal A549 cell line, significantly higher level of MDR1 mRNA and P-gp protein expression was conferred in A549DDP cell lines. In vivo studies were carried out in xenograft tumour-induced female BALB/c-nu/nu mice. Induction of tumours in mice was done with the subcutaneous injection of A549 and A549DDP cells into the selected mice groups. The combinational therapy with RA and DDP has shown a significant reduction in the growth of xenograft tumours. The tumour volume was significantly higher in groups that received RA alone and DDP alone in comparison to the group that received combination therapy. This clearly revealed the synergistic effect of RA and DDP on NSCLC cells. Therefore, the study concluded that stimulation of apoptosis occurred through the down-regulation of Bcl-2 expression, up-regulation of p21, p53 expression, and inhibition of P-gp expression. Treatment with RA and DDP has shown a reduction in the efflux of DDP from NSCLC cells and, hence, caused increased accumulation of DDP in the tumour cell and increased sensitivity to DDP. This proved the resistance reversal activity of RA through inhibition of the MAPK signalling pathway mediated through a c-Jun N-terminal kinase inhibitor [50].

### 3.7. Breast Cancer

Anwar et al. reported the inhibitory effect and binding mechanism of RA to the microtubule affinity-regulating kinase 4 (MARK4). During molecular docking studies, RA has shown high binding affinity (−8.1 kcal/mol) with active sites of MARK4. Further, enzyme inhibition and tau-phosphorylation assays were performed. The results confirmed that RA is a potent inhibitor of MARK4. The cell proliferation and apoptosis studies were carried out on breast cancer cells (MDA-MB-231) and Human adenocarcinoma alveolar basal epithelial cells (A549). These selected cancer cell lines are overexpressed with MARK4, and when they are treated with RA, a differential inhibition of cell growth was observed. Later, RA was applied to the MDA-MB-231 cells at its IC_50_ concentration of 6.204 µM, and after 48 h, immunoblotting studies were performed.

The results confirmed that treatment with RA has decreased the expression of MARK4 in MDA-MB-231 cell lines. Apoptosis studies in MDA-MB-231 cells confirmed the dose-dependent apoptosis effect of RA. These studies suggested that the use of RA as well as its derivatives has potent MARK4 inhibitory activity. Further, RA reduced the overexpression of this protein and thereby controls cancer cell proliferation. It also induces apoptosis [51].

Another study with RA was carried out in triple-negative breast cancer (TNBC) cell lines (MDA-MB-231 and 468 cells). Treatment with RA in both the cell lines inhibited cell growth and further proliferation. These observations confirmed that RA induced cell cycle arrest-related apoptosis and altered the expression of many genes that are related to apoptosis. RA inhibited the G0/G1 phase in MDA-MB-231 cells whereas the S-phase arrest was reported in MDA-MB-468 cells; hence, that apoptotic effect was increased two-fold in MDA-MB-468 cells compared to MDA-MB-231 cells. In addition, there was a significant increase in mRNA expression in MDA-MB-231 cells and MDA-MB-468 cells through the activation of Growth arrest and down-regulated DNA-damage-inducible protein alpha (GADD45A) and BCL-2 interacting protein 3 (BNIP3) genes. It was also observed that RA down-regulated the gene expression of genes belonging to Tumor necrosis factor (TNF) superfamily member 10 (TNFSF10) and baculoviral inhibitor of apoptosis repeat containing 5 (BIRC5) in MDA-MB-468 cells whereas RA reported more than a 4-fold repression of TNF receptor superfamily 11B (TNFRS11B) in MDA-MB-231 cells. Overall this study reported the involvement of several genes related to apoptosis in RA-treated MDA-MB-468 cells in comparison to RA-treated MDA-M-31 cells [52].

Another study was carried out by Juskowiak et al. in MCF-7 breast cancer cell lines. The study confirmed the cytotoxic activity of RA on MCF cell lines and PCR analysis confirmed the involvement of various gene expressions such as ZEB1, MDM2, ABCB1, PTEN, and TWIST1 genes.

The study focussed on the adjunct effect of RA along with doxorubicin (DOX). A significant change in ZEB1 gene has been observed when a combination of RA and DOX was used. Treatment of the MCF-7 cell line with 0.2 µM DOX has shown a 3-fold increase in ZEB1 expression whereas the combination of 0.2 µM DOX with 1.5 µM RA as well as 0.2 µM DOX with 15 µM RA have shown a 9-fold increase in ZEB1 expression. However, 0.2 µM DOX with a higher concentration of RA (50 µM) has shown only a 6.5-fold increase in ZEB1 expression. MDM2 gene expression was found to be decreased at all varying concentrations of RA. During the study, all other genes (ABCB1, PTEN, and TWIST1) examined have shown no significant changes in their expression both with DOX alone or with a combination of DOX and RA [53].

### 3.8. Brain Tumour

The antitumor activity of RA as a *Fyn kinase* inhibitor in glioblastoma multiforme was demonstrated by investigating its effect on cell proliferation, migration, invasion, apoptosis, and gene-protein expression in human giloblastomma cell lines (U251 and U343). A cell counting kit-8 (CCK-8) assay was performed at various concentrations of RA (0, 100, 200, and 400 µM) in U251 and U343 cells. It was reported to exhibit time and dose-dependent inhibition of cell proliferation.

Similar to this observation, immunofluorescence staining also confirmed the dose-dependent inhibition of Fyn expression in U251 and U343 cells. Western blotting studies reported a dose-dependent decrease in the expression of MMP-2 and MMP-9 in U251 and U343 cells. The reduction in expression of MMPs indicated the inhibition of glioma cell invasion and migration. After treatment with RA, an increase in the expression of cleaved caspase-3 and Bax cells was observed. Further, a decrease in the expression of Bcl-2 was observed in both U251 and U343 cells.

In addition to this, the ratio of Bax to Bcl-2 and cleaved caspase-3 to caspase-3 was increased on treatment with RA. This provides an insight into the mechanistic basis for induction of apoptosis in glioma cells. In addition, a decreased protein expression of PI3K, κp-Akt, and NF-κB was observed in selected U251 and U343 cells. This observation suggested the involvement of the PI3K/Akt/NF-B signalling pathway in exerting the antitumor effect of RA in glioma [47].

### 3.9. Osteosarcoma

In 2008, Niforou et al. carried out 2-DE proteomic analysis in osteosarcoma U2OS cells and reported the overexpression of DJ-1 cells, which is also known as protein/nucleic acid deglycase DJ [54]. Similar to this study, Ma et al. focused on proteomic analysis and molecular signalling pathways to evaluate the efficacy of RA against osteosarcoma as well as to clarify the mechanistic pathways involved in antitumour activity [55].

Therefore, an in-depth analysis was carried out to examine the effect of RA on selected osteosarcoma cell lines such as U2OS and MG63. The basic studies for cell viability, apoptosis, cell cycle distribution, migration, and invasion were carried out. Then, various assays such as a CCK-8 assay, wound healing assay, Transwell assay, etc. were performed to trace the signalling molecules involved in the study. Finally, a proteomic analysis confirmed the role of RA as antiproliferative and pro-apoptotic in osteosarcoma cells.

The involvement of both the extrinsic death receptor-mediated pathway and intrinsic mitochondria-mediated apoptotic pathways was confirmed. During the apoptotic process, a higher level of Bax to Bcl ratio, increased production of intracellular reactive oxygen species (ROS), reduced mitochondrial membrane potential, and up-regulation in the cleavage rates of caspase 8, 9, and 3 were observed. Along with anti-proliferation, the inhibition of MMP 2 and 9 expressions resulted in the suppression of migration and invasion of osteosarcoma cells.

After treatment with RA, the up-regulation of p21 and down-regulation of Cdc-25c were observed, and this confirmed that RA induces cell cycle arrest in the G2/M phase. From proteomic analysis, the knockdown of DJ-1 with transfection using shRNA indicated that DJ-1 is the target oncogene biomarker. The overexpression of DJ-1 results in the down-regulation of PTEN protein. Similar to this finding, Wang et al. also reported the down-regulation of DJ-1 in cervical carcinoma which caused a reduction in cell viability and induced apoptosis. The study findings also reported the involvement of the PTEN-PI3K-aKt pathway. The results of molecular level studies performed in RA-treated osteosarcoma cell lines reported a reduction in the level of DJ-1, pPI3K, and p-Akt and an increased level of PTEN. The fluctuation in the protein level indicated that RA-induced apoptosis in osteosarcoma cells through the modulation of the PTEN-PI3kAkt signalling pathway [56].

### 3.10. Gastric Cancer

Han et al. investigated the anti-Warburg effect of RA in gastric carcinoma. Numerous in vitro cell line studies were carried out in human gastric cancer cells (MKN45) followed by in vivo studies in the mouse xenograft model. The dose-dependent cytotoxic effect of RA was observed in MKN 45 cells.

The inhibition in the growth of MKN 45 cells was significant with an IC_50_ value of 240.2 µM. The treatment of MKN45 cells with RA has shown a reduction in glucose uptake and lactate production. These reductions occur due to the inhibition of HIF-1 expression, which affects the glycolytic pathways [34]. Similar observations are reported in CRC cells by Yichum et al. The involvement of interleukin (IL)-6/signal transducer and activator of transcription-2 (STAT-3) pathway in suppressing the Warburg effect was clearly observed from the inhibition of pro-inflammatory cytokines (IL-6) and microRNA (miR-155). Further, in vivo studies were carried out in a xenograft tumour model in which MKN-45 cells are placed subcutaneously in BALB/c-nu mice. The tumour-induced mice were treated with RA at a dose of 2 mg/kg for 14 days and the results demonstrated no loss in body weight, reduction in glucose consumption and lactate production which confirms the inhibitory effect of RA on the Warburg effect [34].

### 3.11. Skin Cancer

A combination of 5 µM of fucoxanthin (FX) and 5 µM of RA labelled as M2 exerted a photo-protective effect. Numerous studies have demonstrated the antioxidant and anti-inflammatory potential of FX and RA as individual compounds. However, the individual compounds have shown the various effects on UV-B radiation-caused DNA damage and other related changes. The study carried out in HaCaT keratinocytes demonstrated that FX being an antioxidant protected DNA from UV-induced damages and, hence, prevented them from cellular apoptosis, whereas, UV-irradiated HaCaT keratinocytes when exposed to RA has reported to reduce the production of reactive oxygen species, prevent DNA damage, and regulate apoptotic markers. However, these individual compounds have shown a significant increase in cell viability and a decrease in reactive oxygen species production, but no effect was observed on the number of apoptotic cells. However, treatment with M2 not only improved cell viability and ROS level but also enhanced cellular production and reduced the number of cells in apoptosis. Apart from these cellular responses, M2 affects cell cycle arrest in both G1 and G2 phases.

Furthermore, pre-treatment with M2 has significantly reduced IL-1β production; down-regulated inflammasome components such as nucleotide-binding domain, leucine-rich-repeat-containing family, pyrin domain-containing 3 (NLRP3), inflammasome adapter protein (ASC), and caspase-1; and also increased the anti-oxidant gene expression of Nrf2 and HO-1 in UV-B irradiated HaCaT cells. These results confirmed the photo-protective effects of M2 which highlighted the down-regulation of NLRP3-inflammasome and up-regulation of the Nrf2 signalling pathway [15].

An in vivo study was carried out in Swiss albino mice, and skin carcinoma was induced by applying 20 µg of 7, 12 di-methylbenz(a) anthracene (DMBA) in 0.1 mL acetone twice weekly for 8 weeks on the skin. Induction of carcinoma was observed by evaluating the alteration of detoxication agents involved in Phase I and Phase II of metabolism, by-products formed during lipid peroxidation, antioxidants, and apoptotic biomarkers (p53, Bcl-2, caspase-3 and caspase-9). After 15 weeks of the oral administration of RA, it was shown to reverse the altered status of the biomarkers that are involved in carcinogenesis induced by DMBA [57].

### 3.12. Renal Carcinoma

The synergistic activity of the hot water extract of *Glechoma hederacea* (HWG) and RA with DDP was evaluated against metastatic renal cancer using renal carcinoma cell lines (RCC 786-O). In addition, the study was conducted on a human renal proximal tubular epithelial cell line (HK-2 cells). An inhibitory effect of varying concentrations of HWG (0, 50, 100, 200 µg/mL), RA (0, 25, 50 and 100 µM) and DDP (5 µM), and DDP was carried out in the RCC 786-O cell lines.

The study reported toxicity on RCC 786-O cells with DDP. Combination of HWG/DDP, as well as RA/DDP, inhibited cell invasion at a concentration of more than 100 µg/mL and more than 25 µM, respectively. The cell cycle analysis in the cancer cell line revealed that the combination treatment of HWG/DDP and RA/DDP caused cytotoxicity by inducing G2/M arrest and induced apoptosis with poly-ADP ribose polymerase (PARP). In vitro studies reported a significant reduction in the expression of phosphorylated focal adhesion kinase (p-FAK) in RCC 786-O cells when RA/DDP combination was applied; however, this was not observed with HWG [58].

### 3.13. Cardiomyocyte

DOX-induced toxicity is related to cardiac dysfunction and heart failure. Screening studies reported the cardioprotective effect of RA. Cell morphological analysis and other experimental studies were carried out on a human cardiomyocyte cell line (AC16) and human induced pluripotent stem-cell derived cardiomyocytes (hiPSC-CMs). The pre-treatment of cell lines with RA at 10 µM has reported suppression of DOX (1µM) induced cell apoptosis and inhibited the activation of caspase-9. The stimulation of heme oxygenase-1 (HO-1), which reduces the production of reactive oxygen species, promoted the expression of various proteins such as histone deacetylase, GATA binding protein 4, and troponin13. These findings suggested the therapeutic benefit of RA with special reference to the cancer therapy-related cardiac dysfunction [59].

### 3.14. Pancreatic Cancer

In vitro and in vivo investigation of anti-tumour activity of RA was carried out in pancreatic cell lines and xenograft nude mice respectively. Among the in vitro studies carried out, MTT assay reported concentration and time-dependent reduction in cell viability in both Panc-1 and SW1990 cell lines. In these cell lines, more than 50% reduction in cell viability was reported at 100 µM. Other in vitro study results emphasized a significant reduction in cell growth, invasion, and migration. RA treatment induced apoptosis in the pancreatic cancer cell line as well as reduced the expression of epithelial-mesenchymal transition markers such as vimentin and N-cadherin.

Further examination confirmed that RA treatment up-regulated the miR-506 level in a concentration-dependent manner in both the cell lines. The results of the luciferase reporter assay confirmed that miR-506 targeted the 3′ untranslated region of MMP-2 and MMP-16. After RA treatment, the suppression of MMP-2 and MMP-16 overexpression resulted in the suppression of invasion and migration of cells. Finally, all these observations were confirmed by in vivo studies on a xenograft nude mice model. Similar to in vitro studies, suppression in pancreatic tumour growth and other related mechanisms was also observed in the tumour tissue transplanted into the nude mice. All these study findings highlighted the mechanistic action of RA as an antitumor agent for pancreatic cancer, which is regulated through miR-506/MMP2/16 axis [60].

### 3.15. Prostate Cancer

Jang et al. carried out a comparative study between RA and suberoylanilide hydroxamic acid (SAHA), a histone deacetylases (HDAC) inhibitor using prostate cancer cell lines PC-3 and DU145. The cell lines treated with RA have been shown to prevent the growth of tumour spheroids and colony formation. In addition, cell proliferation in prostate cancer cell lines PC-3 and DU145 were decreased. Furthermore, both early and late stages of apoptosis were also triggered in PC-3 and DU145 cells which were confirmed by the Annexin V and terminal deoxynucleotidyl transferase dUTP nick end labelling (TUNEL) assays.

Western blot analysis reported inhibition of HDAC2 on the cell lines after treatment with RA as well as SAHA. The study findings also reported the down-regulation of proliferative cell nuclear antigen (PCNA), cyclin D1, and cyclin E1 as well as the up-regulation of p21. A significant increase in apoptosis was reported due to the modulation of Bax, Bd-2, caspase-3, and poly (ADP-ribose) polymerase 1 (PARP-1) and the up-regulation of p53. Altogether, the study emphasized the role of RA in prostate cancer with a special focus to act as an HDAC inhibitor [61].

### 3.16. Hepatocellular Carcinoma

An in vivo study was carried out by Cao et al. using an H22 hepatocarcinoma xenograft tumour model in mice. Intragastric administration of RA was followed in tumour-bearing mice at a dose of 75, 150, and 300 mg/kg once daily for 10 consecutive days. The study focussed on the effect of RA in the NF-κB p65 signalling pathway as well as the molecular mechanism involved during inflammation followed by angiogenesis in a xenograft model. A significant decrease in the elevated level of IL-1β, IL-6, TNF-α, VEGF, and TGF-β was observed in tumour tissue after the treatment with 300 mg/kg of RA. Western blot analysis confirmed a decrease in p65 phosphorylation at all administered doses of RA and qRT-PCR reported a decrease in mRNA level of p65 at 150 mg/kg dose of RA. These reports suggested that RA exerted an anti-tumour mechanism through inhibition of NF-κBp65 and angiogenic factors [62]. Renzulli et al. carried out in vitro cytotoxicity studies in human hepatoma-derived cell lines (Hep G2). RA has shown a significant cytoprotective effect against ochratoxin A and Aflatoxin B1 induced oxidative stress and cell damage. Hep G2 cell lines treated with RA have shown dose-dependent activity resulting in the prevention of DNA fragmentation at higher concentrations and causing inhibition of caspase-3 activation [63].

Another study was carried out on HepG2 cell lines using *Perilla frutescens* leaf extract (PLE) which contained RA. The finding confirmed that PLE-treated HepG2 cells exhibited anti-proliferative activity at a concentration of 105 µg/mL, and a number of apoptosis-related genes are involved in the apoptosis process. Treatment with RA at 10 µg/mL also increased the expression of apoptosis-related genes and apoptosis, but it was less effective in comparison to PLE activities [64].

A study on HepG2 cells treated with a water extract from *Spica prunellae* and RA confirmed a dose-dependent increase in efflux activity of P-glycoprotein; it also increased the multi-drug resistance-associated protein, enhanced the production of ATP intracellularly, activated the translocation of nuclear factor E2-related factor-2 (Nrf2), and enhanced the antioxidant response element-luciferase activity [65].

### 3.17. Leukemia

Similar to the aforementioned study by Cao et al., RA treatment in human leukemia U 937 cells has reported TNF-α induced apoptosis due to the suppression of NF-κB and reactive oxygen species. The inhibition of phosphorylation and degradation of IκBα and the nuclear translocation of p50 and p65 resulted in suppression of NF-κB activation. This suppression of NF-κB also inhibited some of the anti-apoptotic proteins such as IAP-1, IAP-2, and XIAP, whereas RA treatment sensitized TNF-α induced apoptosis and activated caspases [66].

Another study has highlighted the potential role of RA along with all-trans retinoic acid for the treatment of acute promyelocytic leukemia (APL). Flow cytometric analysis was carried out to evaluate the role of each drug in both bone marrow cells of the APL cell line (NB4 cells) and normal bone marrow cells as well as peripheral blood mononuclear cells. The analysis result clearly highlighted that all-trans retinoic acid/RA induced expression of differentiation marker CD11b in the bone marrow of NB4 cells only. It did not induce expression in normal bone marrow cells. This confirmed that RA enhanced all-trans retinoic acid-induced macrophage differentiation in APL patients. The study also emphasized that along with increased expression of CD11b and CD14 cells, the activation of ERK- NFκB occurred, which showed the potential role of RA in the treatment of APL [67].

A combination of ellagic acid and RA with arabinofuranosyl cytosine (AraC) exhibited synergism in exerting anti-proliferative activity in HL-60 promyelocytic leukemia cells. Epigallocatechin gallate, ellagic acid, and RA significantly inhibited the growth of leukemia cells in a dose-dependent manner. These three natural polyphenols have been reported to cause an imbalance in the nuclear deoxyribonucleoside triphosphate (dNTP) and affect DNA synthesis. Co-treatment of epigallocatechin gallate and RA resulted in the attenuation of HL-60 promyelocytic leukemia cells by affecting the G0/G1 phase of the cell cycle, thereby inducing apoptosis [68].

Wu et al. isolated RA from *Salvia miltiorrhiza* and performed an in vitro study on CCRF-CEM and CEM/AD 5000 cells to evaluate the mode of action of RA in acute lymphoblastic U937 cells. Cell morphology and an Annexin V-PI assay have shown dose-dependent inhibition of CCRF-CEM and CEM/AD 5000 cells, which induced apoptosis and necrosis, resulting in cell cycle arrest. The mechanistic analysis reported that RA-induced disruption of the mitochondrial membrane potential activated PARP-cleavage and caspase-independent apoptosis. Molecular docking studies and gene promoter binding motif analyses have shown that it blocked the translocation of p65 from the cytosol to the nucleus and ameliorated cell adhesion to fibronectin, thereby affecting cellular movement [69]. The various multiple targeting mechanisms of RA on various cell lines as explained above is represented in the Figure 5.

## 4. Patents Granted for RA Isolation

Christ et al. filed a patent on the novel technique developed for the isolation of RA from the lemon balm (*Melissa officinalis* Linn.). According to this method, 100 g of grounded balm mint leaves were extracted twice with 2 L of water at 800 °C to 1000 °C for 45 min with stirring. The PH was adjusted between 2 to 2.5 by the addition of 25% hydrochloric acid. The obtained precipitate was centrifuged and filtered. The filtrate was extracted 3 times using di isopropyl ether for 50 mL each and pooled together. The pooled organic phase was evaporated to a residue containing RA of 54% *w*/*w*. The recrystallisation was performed with water [70].

Kott et al. patented the procedure of the isolation of RA from the leaves of Mentha spicata Linn. It includes spearmint (*Mentha spicata*) plants and plant tissues with higher levels of RA. There are also instructions on how to make rosmarinic acid (RA) from spearmint and how to use it as a nutraceutical. In particular, there is a drink made from spearmint plant tissues that has more than 77.5 mg/g of RA per g of dry weight. There are also methods for making this drink and using it to treat an inflammatory or infectious disease. As per our WIPO search, no patents on the novel delivery systems of RA were found, and it was realized that the main challenge in delivering RA as a potential anticancer drug is its poor solubility because of its phenolic nature [71].

## 5. Challenges in Clinical Translation of RA into Anti-Cancer Therapy

Several pharmacokinetic studies using orally administered RA have been published in the literature, showing its low oral bioavailability [72,73,74]. Poor solubility in water and ineffective membrane permeability are two key factors that are linked to the low oral bioavailability of RA bioavailability. RA is an ionisable strong acid with a pH of 2.9. Because of this strong acidic nature, it has pH-dependent distribution coefficient that may have an impact on its permeability or solubility in body fluids [75]. Additionally, research conducted in vitro using Caco-2 cells has shown that RA has a low degree of permeability [76]. In addition, the stability of RA has been identified as a significant contributor to its low bioavailability, particularly after oral administration when passage through the gastrointestinal tract takes place [77].

New technical options for the administration of RA have been researched in light of its poor water solubility, ineffective penetration through biological barriers, high instability, and subsequently limited bioavailability. To cross all said limitation, RA can be associated with or incorporated into cyclodextrins [78], nanoemulsion [79], phospholipid complexes [80], solid lipid nanoparticles [81], and chitosan nanoparticles [82] as part of some techniques. In recent years, NDDS have been utilized to overcome the poor aqueous solubility and gastrointestinal permeability of drugs. However, limited research has been done on novel drug delivery systems (NDDS) loaded with RA against cancer. Those studies have been discussed below.

## 6. Novel Drug Delivery Systems of RA against Cancer

### 6.1. Liposomes

Xue et al. prepared a liposomal nanoformulation of RA. They formulated RA liposomes named as rososomes (RS). The RS were prepared by the classic film hydration method. Phospholipid choline and 1-polmitoyl-2-hydroxy –sn-glycero-3-phosphocholine were taken as lipids and made the RA-lipid matrix (RA-L). The RA-L was prepared by conjugating amphiphilic lipid with RA via esterification. The RA-L was co-assembled with polyethylene glycol and phosphotidyl choline to form RS. The diameter of RS was found to be 198.9 nm and had a poly dispersity index (PDI) of 0.194. They have added ferric iron to the RS for cross linking to stabilize the liposomal structure. The in vivo experiments were conducted in tumor bearing breast cancer mice, which demonstrated that the iron cross linked RS showed significant activity for cancer therapy [83].

Subongkot et al. formulated ultradeformable liposomes (ULs) of RA with fatty acids such as oleic, linoleic, and linolenic acid for enhanced dermal penetration of RA against skin cancer. The size, surface charge, size distribution, shape, % entrapment efficiency (% EE), and % loading efficiency (% LE) of the prepared ULs were all evaluated. The produced ULs with fatty acids had a negative surface charge and had an average particle size between 50.37 ± 0.3 and 59.82 ± 17.3 nm with a size distribution within an acceptable range. Average %EE and %LE percentages were 9 and 24, respectively. According to a study on in vitro skin penetration, RA may be absorbed into the skin much more readily through ULs containing oleic acid than through ULs alone [84].

### 6.2. Polymeric Nanoparticles

Fuster et al. formulated silk fibroin nanoparticles loaded with RA (RA-SFN) against breast cancer (MCF-7) and human cervical carcinoma (HeLa) cell lines. The synthesized RASFNs characteristics were found as a particle diameter of 255 nm and zeta potential of −17 mV with a poly dispersity index of 0.187. About 50% of the total drug content was released in 0.5 h which indicated a rapid release in physiological conditions. Cell line studies revealed that the concentration dependent cancer cell death was observed after treatment with RA-SNs, which is not observed up to the duration of 48 h in unloaded SFNs. The cell death in MCF-7 and HeLa was evident from the IC_50_ values of 1.568 and 1.377 mg/mL, respectively. Further, the inhibition of cell proliferation was observed in cancer cell lines after cell cycle and apoptosis studies [85].

Tabatabaein et al. formulated N-doped carbon dots (N-CND) loaded with RA by a hydrothermal method against cancer. The particle size range of N-CND was 35 ± 10 nm and the zeta potential was found to be −3.6 mV. An MTT assay was performed to study the apoptotic effect, which was found to be increased with an increase in the concentration of N-CND from 25 to 150 µg/mL. Therefore, the novel carrier of RA was considered to be a promising and cost effective carrier for drug delivery in cancer treatment [86].

Ching et al. formulated PEGylated RA-derived nanoparticles (RANPs) for the treatment of inflammatory bowel disease and colorectal cancer. PEGylated RA was produced from RA and a PEG-containing amine in a single step, and it then self-assembled in a buffer to produce nanoparticles (RANPs) with a diameter of 63.5 ± 4.0 nm. In physiological media, the resultant RANPs displayed good colloidal stability for up to two weeks. RANPs could effectively scavenge hydrogen peroxide, preventing cells from suffering harm from hydrogen peroxide-induced oxidative stress. In the inflamed colon, RANPs reduced the expression and production of common pro-inflammatory cytokines [87].

### 6.3. Gold Nanoparticles

Gancalves et al. formulated gold nanoparticles (AuNP) of RA against breast cancer. The AuNP were prepared using a gold chloride trihydrate solution (1 mM), silver nitrate (1 mM), and L-ascorbic acid (2 mM) at room temperature under magnetic stirring of 800 rpm. The formed AuNPs were stored for 2 h at 2 °C and centrifuged at 1500 rpm for 20 min to remove unreacted reagents. The particle diameter of AuNPs ranged from 151.1 nm to 175 nm, and it had a poly dispersity index of 0.240–0.337 and zeta potential of −15.7 mV to −22.6 mV. The morphological characterization revealed that the AuNPs were in a spherical or quasi spherical shape and in other planar structures. The TEM studies revealed the surface roughness on particles. The particles show successful internalization of RA into murine fibroblasts and human breast cancer cells [88].

### 6.4. Solid Lipid Nanoparticles (SLNs)

Campos et al. formulated SLNs of RA with Witepsol as lipid and tween 80 as surfactant using a hot melt extrusion method. They have used 1%, 2%, and 3% *v*/*v* of lipids and 0.5%, 1%, and 1.5% *w*/*v* of lipids. The results showed that Witepsol H15 generated nanoparticles with initial mean diameters between 270 and 1000 nm. While monitoring the mean particle size of various formulations using photon correlation spectroscopy, the durability of the nanoparticle systems was assessed over the course of 28 days in aqueous solution held at refrigerator temperature (about 5 °C). Thermal studies of the nanoparticles with differential scanning alorimetry (DSC) and Fourier transform infrared spectroscopy (FTIR) were carried out to establish RA entrapment. To quantitatively evaluate the RA in the supernatants, the association efficiencies percentages (AE%) were calculated using HPLC [89].

### 6.5. Silver Nanoparticles

Bhatt et al. formulated silver nanoparticles with rosmarinic acid caps (Ro-AgNPs), which were used as a probe for selective colorimetric detection of cyanide (CN-) and chromium(VI) [Cr(VI)] in aqueous solutions under various conditions. In its interactions with the AgNPs, the carbon atom of CN- transfers electrons from the HOMO to the open orbitals of the coordinatively unsaturated surface atom (Ag^0^). For CN- and Cr(VI), the limits of detection were discovered to be 0.01 and 0.03 M, respectively. The material was also employed to detect CN- and Cr(VI) in actual samples, with positive findings. The immobilisation of nanoparticles onto the agarose film allowed for the preparation of agarose-based strips for field application, which were successfully used for the detection of CN- and Cr(VI) in water [90].

The overall summary of the study in the form of a graphical image is shown in Figure 6.

## 7. Conclusions

Bioactive substances of herbal origin have been used from olden days for the treatment of various diseases. Cancer is a life-threatening disease with several limitations and side effects associated with its treatment. Herbal substances made scientists focus on their significance in the treatment of various types of cancers. RA is one such phytoconstituents that has been reported by several authors for its anticancer activity with minimal side effects. In this comprehensive review, the prevalence of various types of cancers across the globe, various extraction processes of RA, the mechanism of the action of RA against various cancers including in vivo and in vitro cell line studies have been discussed in detail. Further, the review outlines the problems encountered during the clinical translation of RA into therapeutic delivery systems loaded with RA against cancer.

Poor solubility in water and ineffective membrane permeability are two major factors which are linked to low RA’s bioavailability. Its ionization at acidic pH and pH-dependent distribution coefficient poses untoward impact on its permeability or solubility in body fluids. Looking at these pit falls, new technical options for the administration of RA have been researched in light of its poor water solubility, ineffective penetration through biological barriers, high instability, and subsequently limited bioavailability. To cross all said limitations, there are reports wherein RA has been incorporated into cyclodextrins, nanoemulsion, phospholipid complexes, solid lipid nanoparticles, and chitosan nanoparticles to improve its anticancer efficacy. These approaches have provided significant improvement in overcoming the oral bioavailability of RA as well as proven to treat cancer in a much lower dose. Despite this success, there are limited studies related to the development of NDDS.

The stability of nanoparticles, laborious and time-consuming manufacturing techniques, inadequate product yield, inadequate drug loading, clinical toxicity of nanoparticles, and difficulties in scaling are the main barriers to achieving success with RA loaded nanoformulations at a commercial scale. However, there are some flavonoids found in plants, like curcumin, for which nanoparticles have been created, successfully undergone clinical testing, and are currently offered for sale for commercial use. To enable the creation of RA-loaded NDDS, it is crucial to comprehend and optimize the quality target product profile, key formulation, and processing characteristics, as well as to assess the safety and efficacy profile of the formulation post-scaling up. Furthermore, a comparison of long-term stability data and accelerated stability for the developed formulation would further help in understanding and establishing a good stability profile of formulation. Establishment of good correlation between in vitro and in vivo performance of formulation such as pharmacokinetic as well as pharmacodynamic interactions would enable better clinical application of the developed nanoformulations.

## Figures and Tables

**Figure 1 pharmaceutics-14-02401-f001:**
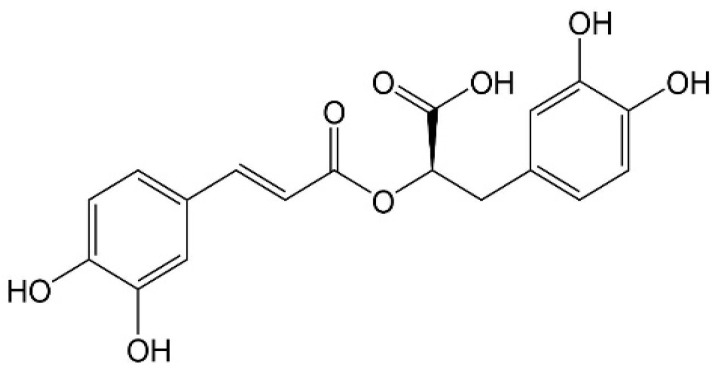
Chemical structure of Rosmarinic acid (RA).

**Figure 2 pharmaceutics-14-02401-f002:**
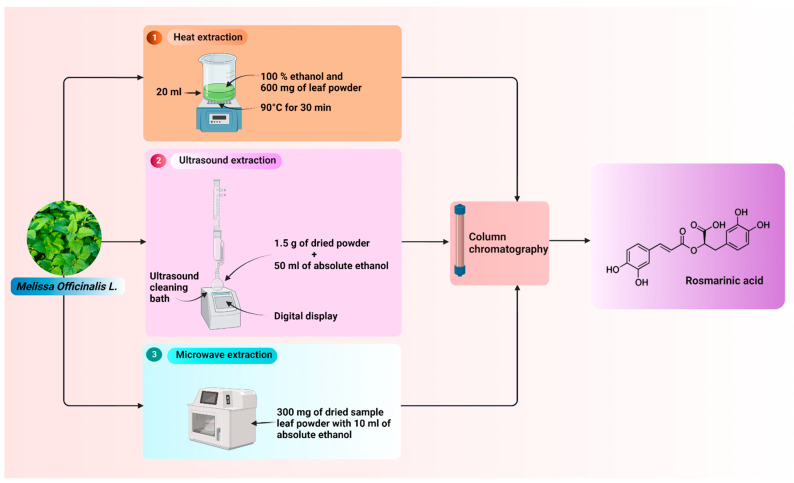
Various extraction and isolation methods of RA.

**Figure 3 pharmaceutics-14-02401-f003:**
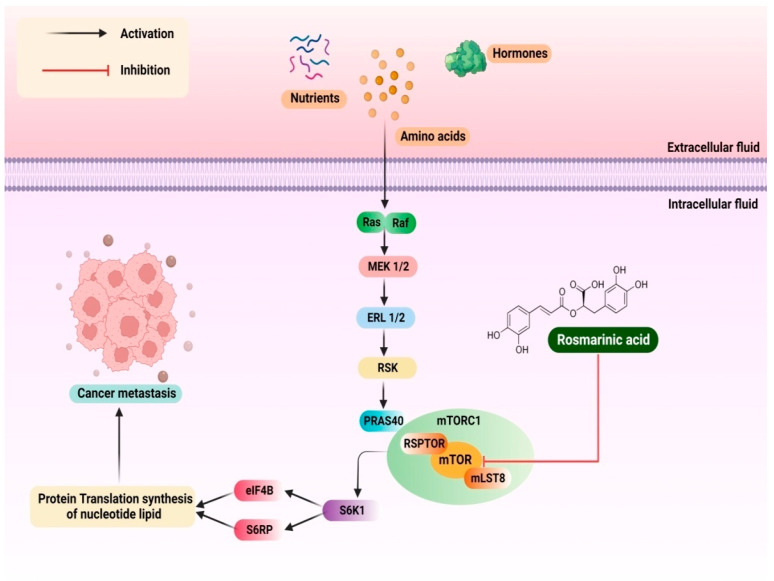
Figure depicting mechanism of rosmarinic acid inhibition of mTOR/S6K1 in cervical cancer cell lines (Henrietta Lacks) by RA.

**Figure 4 pharmaceutics-14-02401-f004:**
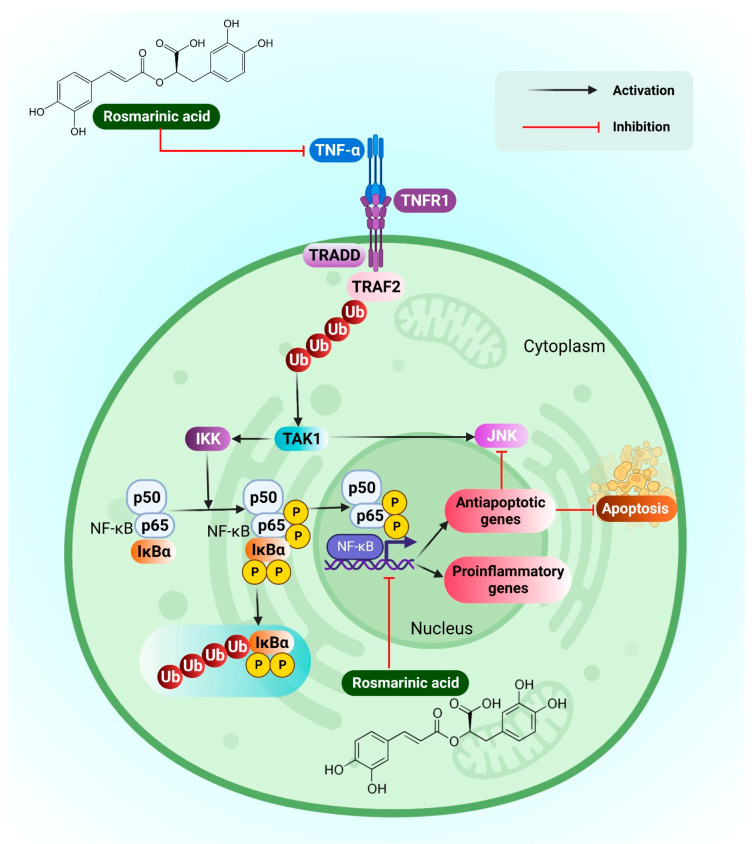
Figure depicting molecular mechanism of rosmarinic acidon A 549 (Lung cancer cell lines) through inhibition of NF-kβ.

**Figure 5 pharmaceutics-14-02401-f005:**
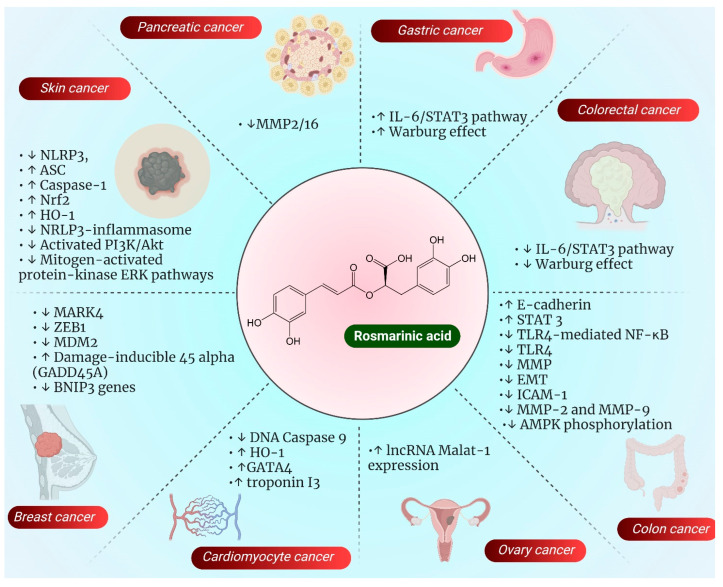
Figure depicting pharmacological action of rosmarinic acid on various molecular targets leading to different types of cancer based on the studies performed on various cancer cell lines. ↑ indicates upregulation and symbol ↓ indicates downregulation.

**Figure 6 pharmaceutics-14-02401-f006:**
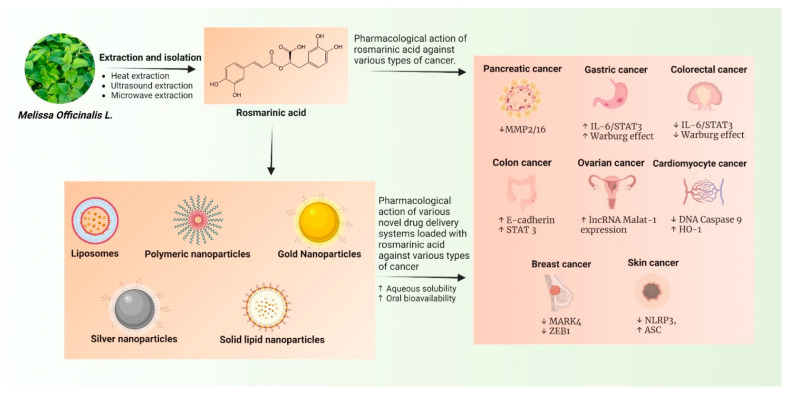
Schematic representation of entire flow diagram of review highlighting the process of extraction, isolation, and anticancer activities of naïve rosmarinic acid and its nanocarriers. ↑ indicates upregulation and symbol ↓ indicates downregulation.

**Table 1 pharmaceutics-14-02401-t001:** Various types of cancers with estimated number of new cases and deaths in 2022 in US [4].

S.No.	Type of Cancer	Estimated New Cases	Estimated Deaths
1	Breast	287,850 (Female)	43,250
		2710 (male)	530
2.	Colon	106,180	52,580
3.	Rectum	44,850	
4.	Kidney	79,000	13,920
5.	Leukaemia	60,650	24,000
6.	Liver	41,260	30,520
7.	Lung and bronchus	236,740	130,180
8.	Lymphoma	89,010	21,170
9.	Pharynx	54,000	11,230
10.	Ovary	19,880	12,810
11.	Prostate	268,490	34,500

## Data Availability

Not applicable.

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
