# Peer review of "Journey of Rosmarinic Acid as Biomedicine to Nano-Biomedicine for Treating Cancer: Current Strategies and Future Perspectives"

_pharmaceutics, 2022, doi:10.3390/pharmaceutics14112401_

Round 1

Reviewer 1 Report

The manuscript "Journey of rosmarinic acid as biomedicine to nano-biomedicine for treating cancer: current strategies and future perspectives" is a comprehensive overview of the prevalence of various types of cancer throughout the world, the various processes for isolating rosmarinic acid and the mechanisms of its action on various types of cancer are given. The authors in their review describe therapeutic delivery systems with rosmarinic acid. I believe that the manuscript may be accept in present form.

Author Response

Comment 1: The manuscript "Journey of rosmarinic acid as biomedicine to nano-biomedicine for treating cancer: current strategies and future perspectives" is a comprehensive overview of the prevalence of various types of cancer throughout the world, the various processes for isolating rosmarinic acid and the mechanisms of its action on various types of cancer are given. The authors in their review describe therapeutic delivery systems with rosmarinic acid. I believe that the manuscript may be accept in present form.

Response: Authors are thankful to the learned reviewer for accepting the manuscript.

Reviewer 2 Report

It was a review paper about the application of rosmarinic acid against different types of cancer. Here are some comments on this study that should be considered before publication:

 1.       The type of your manuscript is review not article, please correct it.

2.       There are lots of typos- and grammatical mistakes in the text that should be corrected.

3.       Please refer to Table 1 in the main text.

4.       Please explain more about the figure caption of figure 3. The same for figures 4 and 5.

5.       Please add more samples from other types of nanocarriers used for the delivery of rosmarinic acid.

6.       Please add future perspectives related to rosmarinic acid.

Author Response

1. The type of your manuscript is review not article, please correct it.

Response: Corrected as per the suggestion.

2. There are lots of typos- and grammatical mistakes in the text that should be corrected.

Response: Corrected as per the suggestion.

3. Please refer to Table 1 in the main text.

Response: Corrected as per the suggestion.

4. Please explain more about the figure caption of figure 3. The same for figures 4 and 5.

Response: As per the valuable suggestions of the learned reviewer, we have edited the captions for figures 3, 4 and 5.

5. Please add more samples from other types of nanocarriers used for the delivery of rosmarinic acid.

Response: Many thanks for this worthy suggestions. We went to various databases such as Google Scholar, Science Direct and PubMed and added the studies of all the possible nanocarrier of rosmarinic acid that have been formulated till date for treatment of cancer. The newly added nanocarriers are listed in section 6.

6. Please add future perspectives related to rosmarinic acid.

Response: As per the suggestions, we have added future perspectives

Reviewer 3 Report

The review article entitled "Journey of rosmarinic acid as biomedicine to nano-biomedicine for treating cancer: current strategies and future perspectives" is a well-written maniscript which presents an interesting review of the use of rosmarinic acid in various types of cancer. 

It is known that cancer is a complex, multipotent pathology that evolves in different ways. Using the term "treatment" of cancer with plant metabolites is a bit risky, but there is a potential to study the bioavailability of active metabolites in various cancer cell lines. 

There are some minor revisions to check in yellow (file attached). 

Author Response

The review article entitled "Journey of rosmarinic acid as biomedicine to nano-biomedicine for treating cancer: current strategies and future perspectives" is a well-written maniscript which presents an interesting review of the use of rosmarinic acid in various types of cancer. It is known that cancer is a complex, multipotent pathology that evolves in different ways. Using the term "treatment" of cancer with plant metabolites is a bit risky, but there is a potential to study the bioavailability of active metabolites in various cancer cell lines. There are some minor revisions to check in yellow (file attached). 

Response: Authors are really thankful and appreciate the learned reviewer for putting his/her effort in suggesting notable points for improving the quality of manuscript. We have incorporated all the suggested changes in the manuscript.

Reviewer 4 Report

The manuscript is an interesting review about rosmarinic acid as anticancer phytochemical and novel drug delivery approaches pertaining to its anticancer activity.

This is a relevant topic that falls within the aims and scope of the journal.

The paper is well written and well structured. Just two comments:

The abstract needs to be revised (200 words max)

A useful index and a graphical summary should be added to the manuscript.

Author Response

1. The manuscript is an interesting review about rosmarinic acid as anticancer phytochemical and novel drug delivery approaches pertaining to its anticancer activity. This is a relevant topic that falls within the aims and scope of the journal. The paper is well written and well structured. 

Response: Authors are thankful to the learned reviewer for appreciating the manuscript.

2. The abstract needs to be revised (200 words max)

Response: Done as suggested

3. A useful index and a graphical summary should be added to the manuscript.

Response: We have provided it as Fig.6.